# Fast Structured Decoding for Sequence Models

**Zhiqing Sun**[1,*] **Zhuohan Li**[2,*] **Haoqing Wang**[3] **Di He**[3] **Zi Lin**[3] **Zhi-Hong Deng**[3]
[1]Carnegie Mellon University  [2]University of California, Berkeley  [3]Peking University
zhiqings@cs.cmu.edu  zhuohan@cs.berkeley.edu
{wanghaoqing,di_he,zi.lin,zhdeng}@pku.edu.cn

## Abstract

Autoregressive sequence models achieve state-of-the-art performance in domains like machine translation. However, due to the autoregressive factorization nature, these models suffer from heavy latency during inference. Recently, non-autoregressive sequence models were proposed to reduce the inference time. However, these models assume that the decoding process of each token is conditionally independent of others. Such a generation process sometimes makes the output sentence inconsistent, and thus the learned non-autoregressive models could only achieve inferior accuracy compared to their autoregressive counterparts. To improve the decoding consistency and reduce the inference cost at the same time, we propose to incorporate a structured inference module into the non-autoregressive models. Specifically, we design an efficient approximation for Conditional Random Fields (CRF) for non-autoregressive sequence models, and further propose a dynamic transition technique to model positional contexts in the CRF. Experiments in machine translation show that while increasing little latency (8∼14*ms*), our model could achieve significantly better translation performance than previous non-autoregressive models on different translation datasets. In particular, for the WMT14 En-De dataset, our model obtains a BLEU score of 26.80, which largely outperforms the previous non-autoregressive baselines and is only 0.61 lower in BLEU than purely autoregressive models.[1]

## 1 Introduction

Autoregressive sequence models achieve great success in domains like machine translation and have been deployed in real applications [1, 2, 3, 4, 5]. However, these models suffer from high inference latency [1, 2], which is sometimes unaffordable for real-time industrial applications. This is mainly attributed to the autoregressive factorization nature of the models: Considering a general conditional sequence generation framework, given a context sequence $x = (x_1, ..., x_T)$ and a target sequence $y = (y_1, ..., y_{T'})$, autoregressive sequence models are based on a chain of conditional probabilities with a left-to-right causal structure:

$$p(y|x) = \prod_{i=1}^{T'} p(y_i|y_{<i}, x),\qquad(1)$$

where $y_{<i}$ represents the tokens before the $i$-th token of target $y$. See Figure 1(a) for the illustration of a state-of-the-art autoregressive sequence model, Transformer [1]. The autoregressive factorization makes the inference process hard to be parallelized as the results are generated token by token sequentially.

Table 1: Cases on IWSLT14 De-En. Compared to their ART counterparts, NART models suffer from severe decoding inconsistency problem, which can be solved by CRF-based structured decoding.

| | |
|---|---|
| Source: | jeden morgen fliegen sie 240 kilometer zur farm . |
| Target: | every morning , they fly 240 miles into the farm . |
| ART: | every morning , they fly 240 miles to the farm . |
| NART: | every morning , you fly 240 miles to every morning . |
| NART-CRF: | every morning , they fly 240 miles to the farm . |
| Source: | ich weiß , dass wir es können , und soweit es mich betrifft ist das etwas , was die welt jetzt braucht . |
| Target: | i know that we can , and as far as i 'm concerned , that 's something the world needs right now . |
| ART: | i know that we can , and as far as i 'm concerned , that 's something that the world needs now |
| NART: | i know that we can it , , as as as as it it it is , it 's something that the world needs now . |
| NART-CRF: | i know we can do it , and as far as i 'm concerned that 's something that the world needs now . |

Recently, non-autoregressive sequence models [6, 7, 8, 9] were proposed to alleviate the inference latency by removing the sequential dependencies within the target sentence. Those models also use the general encoder-decoder framework: the encoder takes the context sentence $x$ as input to generate contextual embedding and predict the target length $T'$, and the decoder uses the contextual embedding to predict each target token:

$$p(y|x) = p(T'|x) \cdot \prod_{i=1}^{T'} p(y_i|x) \qquad (2)$$

The non-autoregressive sequence models take full advantage of parallelism and significantly improve the inference speed. However, they usually cannot get results as good as their autoregressive counterparts. As shown in Table 1, on the machine translation task, compared to AutoRegressive Translation (ART) models, Non-AutoRegressive Translation (NART) models suffer from severe decoding inconsistency problem. In non-autoregressive sequence models, each token in the target sentence is generated independently. Thus the decoding consistency (e.g., word co-occurrence) cannot be guaranteed on the target side. The primary phenomenon that can be observed is the multimodality problem: the non-autoregressive models cannot model the highly multimodal distribution of target sequences properly [6]. For example, an English sentence "Thank you." can have many correct German translations like "Danke.", "Danke schon.", or "Vielen Dank.". In practice, this will lead to inconsistent outputs such as "Danke Dank." or "Vielen schon.".

To tackle this problem, in this paper, we propose to incorporate a structured inference module in the non-autoregressive decoder to directly model the multimodal distribution of target sequences. Specifically, we regard sequence generation (e.g., machine translation) as a sequence labeling problem and propose to use linear-chain Conditional Random Fields (CRF) [10] to model richer structural dependencies. By modeling the co-occurrence relationship between adjacent words, the CRF-based structured inference module can significantly improve decoding consistency in the target side. Different from the probability product form of Equation 2, the probability of the target sentence is globally normalized:

$$p(y|x) = p(T'|x) \cdot \mathrm{softmax}\left(\sum_{i=2}^{T'} \theta_{i-1,i}(y_{i-1}, y_i)\Big|x\right) \qquad (3)$$

where $\theta_{i-1,i}$ is the pairwise potential for $y_{i-1}$ and $y_i$. Such a probability form could better model the multiple modes in target translations.

However, the label size (vocabulary size) used in typical sequence models is very large (e.g., 32k) and intractable for traditional CRFs. Therefore, we design two effective approximation methods for the CRF: low-rank approximation and beam approximation. Moreover, to leverage the rich contextual information from the hidden states of non-autoregressive decoder and to improve the expressive power of the structured inference module, we further propose a dynamic transition technique to model positional contexts in CRF.

We evaluate the proposed end-to-end model on three widely used machine translation tasks: WMT14 English-to-German/German-to-English (En-De/De-En) tasks and IWSLT14 German-to-English

task. Experimental results show that while losing little speed, our NART-CRF model could achieve significantly better translation performance than previous NART models on several tasks. In particular, for the WMT14 En-De and De-En tasks, our model obtains BLEU scores of 26.80 and 30.04, respectively, which largely outperform previous non-autoregressive baselines and are even comparable to the autoregressive counterparts.

## 2 Related Work

### 2.1 Non-autoregressive neural machine translation

Non-AutoRegressive neural machine Translation (NART) models aim to speed up the inference process for real-time machine translation [6], but their performance is considerably worse than their ART counterparts. Most previous works attributed the poor performance to unavoidable conditional independence when predicting each target token, and proposed various methods to solve this issue.

Some methods alleviated the multimodality phenomenon in vanilla NART training: [6] introduced the sentence-level knowledge distillation [11, 12] to reduce the multimodality in the raw data; [8] designed two auxiliary regularization terms in training; [7] proposed to leverage hints from the ART models to guide NART's attention and hidden states. Our approach is orthogonal to these training techniques. Perhaps the most similar one to our approach is [13], which introduced the Connectionist Temporal Classification (CTC) loss in NART training. Both CTC and CRF can reduce the multimodality effect in training. However, CTC can only model a unimodal target distribution, while CRF can model a multimodal target distribution effectively.

Other methods attempted to model the multimodal target distribution by well-designed hidden variables as decoder input: [6] introduced the concept of fertilities from statistical machine translation models [14] into the NART models; [9] used an iterative refinement process in the decoding process of their proposed model; [15] and [16] embedded an autoregressive sub-module that consists of discrete latent variables into their models. In comparison, our NART-CRF models use a simple design of decoder input, but model a richer structural dependency for the decoder output.

### 2.2 Structured learning for machine translation

The idea of recasting the machine translation problem as a sequence labeling task can be traced back to [17], where a CRF-based method was proposed for Statistical Machine Translation (SMT). They simplify the CRF training by (1) limiting the possible "labels" to those that are observed during training and (2) enforcing sparsity in the model. In comparison, our proposed low-rank approximation and beam approximation are more suitable for neural network models.

Structured prediction provides a declarative language for specifying prior knowledge and structural relationships in the data [18]. Our approach is also related to other works on structured neural sequence modeling. [19] neuralizes an unsupervised Hidden Markov Model (HMM). [20] proposed to incorporate richer structural distribution for the attention mechanism. They both focus on the internal structural dependencies in their models, while in this paper, we directly model richer structural dependencies for the decoder output.

Finally, our work is also related to previous work on combining neural networks with CRF for sequence labeling. [21] proposed a unified neural network architecture for sequence labeling. [22] proposed a globally normalized transition-based neural network on a task-specific transition system.

## 3 Fast Structured Decoding for Sequence Models

In this section, we describe the proposed model in the context of machine translation and use "source" and "context" interchangeably. The proposed NART-CRF model formulates non-autoregressive translation as a sequence labeling problem and use Conditional Random Fields (CRF) to solve it. We first briefly introduce the Transformer-based NART architecture and then describe the CRF-based structured inference module. Figure 1(b) illustrates our NART-CRF model structure.

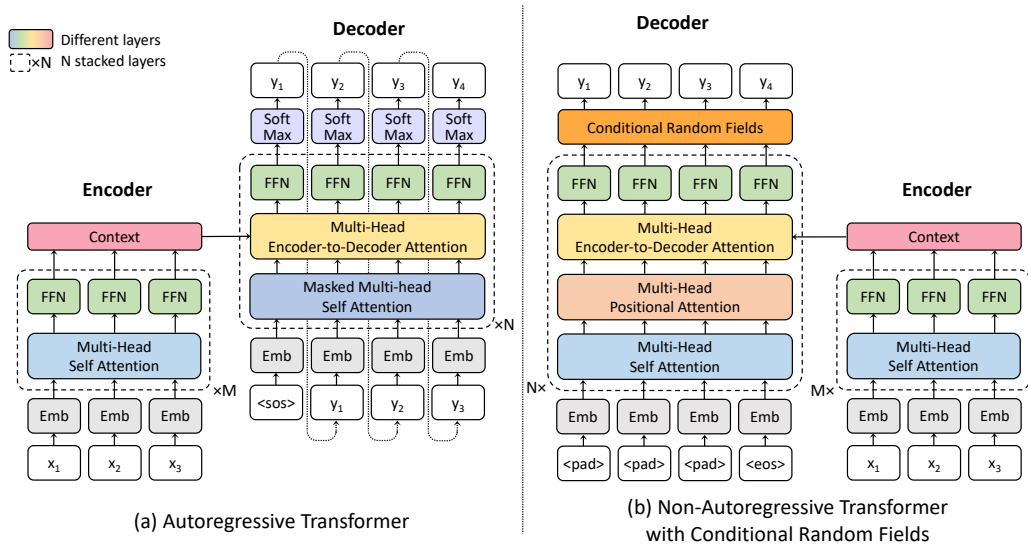

Figure 1: Illustration of the Transformer model and our Transformer-based NART-CRF model

## 3.1 Transformer-based Non-autoregressive Translation Model

The model design follows the Transformer architecture [1] with an additional positional attention layer proposed by [6]. We refer the readers to [1, 6, 23] for more details about the model.

**Encoder-decoder framework**   Non-autoregressive machine translation can also be formulated in an encoder-decoder framework [3]. Same as the ART models, the encoder of NART models takes the embeddings of source tokens as inputs and generates the context representation. However, as shown in Equation 2, the NART decoder does not use the autoregressive factorization, but decodes each target token independently given the target length $T'$ and decoder input $z$.

Different from the previous work [6], we choose a quite simple design of $z$, i.e., we use a sequence of padding symbols $\langle pad \rangle$ followed by an end-of-sentence symbol $\langle eos \rangle$ as the decoder input $z$. This design simplifies the decoder input to the most degree but shows to work well in our experiment.

**Multi-head attention**   ART and NART Transformer models share two types of multi-head attentions: multi-head self-attention and multi-head encoder-to-decoder attention. The NART model additionally uses multi-head positional attention to model local word orders within the sentence [6]. A general attention mechanism can be formulated as the weighted sum of the value vectors $V$ using query vectors $Q$ and key vectors $K$:

$$\text{Attention}(Q, K, V) = \text{softmax}\left(\frac{QK^T}{\sqrt{d_{model}}}\right) \cdot V, \tag{4}$$

where $d_{model}$ represents the dimension of hidden representations. For self-attention, $Q$, $K$ and $V$ are hidden representations of the previous layer. For encoder-to-decoder attention, $Q$ refers to hidden representations of the previous layer, whereas $K$ and $V$ are context vectors from the encoder. For positional attention, positional embedding is used as $Q$ and $K$, and hidden representations of the previous layer are used as $V$.

The position-wise Feed-Forward Network (FFN) is applied after multi-head attentions in both encoder and decoder. It consists of a two-layer linear transformation with ReLU activation:

$$\text{FFN}(x) = \max(0, xW_1 + b_1)W_2 + b_2 \tag{5}$$

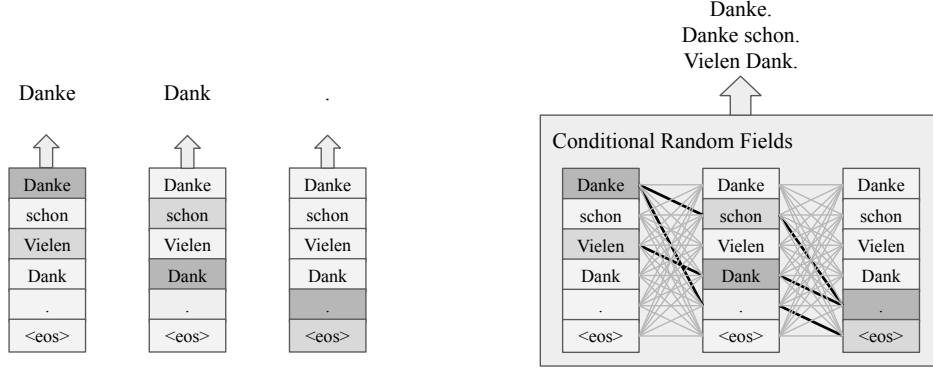

Figure 2: Illustration of the decoding inconsistency problem in non-autoregressive decoding and how a CRF-based structured inference module solves it.

## 3.2 Structured inference module

In this paper, we propose to incorporate a structured inference module in the decoder part to directly model multimodality in NART models. Figure 2 shows how a CRF-based structured inference module works. In principle, this module can be any structured prediction model such as Conditional Random Fields (CRF) [10] or Maximum Entropy Markov Model (MEMM) [24]. Here we focus on linear-chain CRF, which is the most widely applied model in the sequence labeling literature. In the context of machine translation, we use "label" and "token" (vocabulary) interchangeably for the decoder output.

**Conditional random fields**  CRF is a framework for building probabilistic models to segment and label sequence data. Given the sequence data $x = (x_1, \cdots, x_n)$ and the corresponding label sequence $y = (y_1, \cdots, y_n)$, the likelihood of $y$ given $x$ is defined as:

$$P(y|x) = \frac{1}{Z(x)} \exp \left( \sum_{i=1}^{n} s(y_i, x, i) + \sum_{i=2}^{n} t(y_{i-1}, y_i, x, i) \right) \tag{6}$$

where $Z(x)$ is the normalizing factor, $s(y_i, x, i)$ is the label score of $y_i$ at the position $i$, and $t(y_{i-1}, y_i, x, i)$ is the transition score from $y_{i-1}$ to $y_i$. The CRF module can be end-to-end jointly trained with neural networks using negative log-likelihood loss $\mathcal{L}_{CRF} = -\log P(y|x)$. Note that when omitting the transition score $t(y_{i-1}, y_i, x, i)$, Equation 6 is the same as vanilla non-autoregressive models (Equation 2).

**Incorporating CRF into NART model**  For the label score, a linear transformation of the NART decoder's output $h_i$: $s(y_i, x, i) = (h_i W + b)_{y_i}$ works well, where $W \in \mathbb{R}^{d_{model} \times |\mathcal{V}|}$ and $b \in \mathbb{R}^{|\mathcal{V}|}$ are the weights and bias of the linear transformation. However, for the transition score, naive methods require a $|\mathcal{V}| \times |\mathcal{V}|$ matrix to model $t(y_{i-1}, y_i, x, i) = M_{y_{i-1}, y_i}$. Also, according to the widely-used forward-backward algorithm [10], the likelihood computation and decoding process requires $O(n|\mathcal{V}|^2)$ complexity through dynamic programming [10, 25, 26], which is infeasible for practical usage (e.g. a 32k vocabulary).

**Low-rank approximation for transition matrix**  A solution for the above issue is to use a low-rank matrix to approximate the full-rank transition matrix. In particular, we introduce two transition embedding $E_1, E_2 \in \mathbb{R}^{|\mathcal{V}| \times d_t}$ to approximate the transition matrix:

$$M = E_1 E_2^T \tag{7}$$

where $d_t$ is the dimension of the transition embedding.

**Beam approximation for CRF**  Low-rank approximation allows us to calculate the unnormalized term in Equation 6 efficiently. However, due to numerical accuracy issues[2], both the normalizing

factor $Z(x)$ and the decoding process require the full transition matrix, which is still unaffordable. Therefore, we further propose beam approximation to make CRF tractable for NART models.

In particular, for each position $i$, we heuristically truncate all $|\mathcal{V}|$ candidates to a pre-defined beam size $k$. We keep $k$ candidates with highest label scores $s(\cdot, x, i)$ for each position $i$, and accordingly crop the transition matrix between each pair of $i-1$ and $i$. The forward-backward algorithm is then applied on the truncated beam to get either normalizing factor or decoding result. In this way, the time complexities of them are reduced from $O(n|\mathcal{V}|^2)$ to $O(nk^2)$ (e.g., for the normalizing factor, instead of a sum over $|\mathcal{V}|^n$ possible paths, we sum over $k^n$ paths in the beam). Besides, when calculating the normalizing factor for a training pair $(x, y)$, we explicitly include each $y_i$ in the beam to ensure that the approximated normalizing factor $Z(x)$ is larger than the unnormalized path score of $y$.

The intuition of the beam approximation is that for the normalizing factor, the sum of path scores in such a beam (the approximated $Z(x)$) is able to predominate the actually value of $Z(x)$, while it is also reasonable to assume that the beam includes each label of the best path $y^*$.

**Dynamic CRF transition**   In the traditional definition, the transition matrix $M$ is fixed for each position $i$. A dynamic transition matrix that depends on the positional context could improve the representation power of CRF. Here we use a simple but effective way to get a dynamic transition matrix by inserting a dynamic matrix between the product of transition embedding $E_1$ and $E_2$:

$$M^i_{dynamic} = f([h_{i-1}, h_i]), \tag{8}$$

$$M^i = E_1 M^i_{dynamic} E_2^T, \tag{9}$$

$$t(y_{i-1}, y_i, x, i) = M^i_{y_{i-1}, y_i}, \tag{10}$$

where $[h_{i-1}, h_i]$ is the concatenation of two adjacent decoder outputs, and $f : \mathbb{R}^{2d_{model}} \rightarrow \mathbb{R}^{d_t \times d_t}$ is a two-layer Feed-Forward Network (FFN).

**Latency of CRF decoding**   Unlike vanilla non-autoregressive decoding, the CRF decoding can no longer be parallelized. However, due to our beam approximation, the computation of linear-chain CRF $O(nk^2)$ is in theory still much faster than autoregressive decoding. As shown in Table 2, in practice, the overhead is only $8 \sim 14ms$.

**Exact Decoding for Machine Translation**   Despite fast decoding, another promise of this approach is that it provides an exact decoding framework for machine translation, while the de facto standard beam search algorithm for ART models cannot provide such guarantee. CRF-based structured inference module can solve the label bias problem [10], while locally normalized models (e.g. beam search) often have a very weak ability to revise earlier decisions [22].

**Joint training with vanilla non-autoregressive loss**   In practice, we find that it is beneficial to include the original NART loss to help the training of the NART-CRF model. Therefore, our final training loss $\mathcal{L}$ is a weighted sum of the CRF negative log-likelihood loss (Equation 3) and the Non-AutoRegressive (NAR) negative log-likelihood loss (Equation 2):

$$\mathcal{L} = \mathcal{L}_{CRF} + \lambda \mathcal{L}_{NAR}, \tag{11}$$

where $\lambda$ is the hyperparameter controlling the weight of different loss terms.

## 4   Experiments

### 4.1   Experimental settings

We use several widely adopted benchmark tasks to evaluate the effectiveness of our proposed models: IWSLT14[3] German-to-English translation (IWSLT14 De-En) and WMT14[4] English-to-German/German-to-English translation (WMT14 En-De/De-En). For the WMT14 dataset, we use Newstest2014 as test data and Newstest2013 as validation data. For each dataset, we split word tokens into subword units following [2], forming a 32k word-piece vocabulary shared by source and target languages.

Table 2: Performance of BLEU score on WMT14 En-De/De-En and IWSLT14 De-En tasks. The number in the parentheses denotes the performance gap between NART models and their ART teachers. "/" denotes that the results are not reported. LSTM-based results are from [2, 27]; CNN-based results are from [5, 28]; Transformer [1] results are based on our own reproduction.[6]

| Models | WMT14 En-De | WMT14 De-En | IWSLT14 De-En | Latency | Speedup |
|---|---|---|---|---|---|
| *Autoregressive models* | | | | | |
| LSTM-based [2] | 24.60 | / | 28.53 | / | / |
| CNN-based [5] | 26.43 | / | 32.84 | / | / |
| Transformer [1] (beam size = 4) | 27.41 | 31.29 | 33.26 | $387ms^{\ddagger}$ | $1.00\times$ |
| *Non-autoregressive models* | | | | | |
| FT [6] | 17.69 (5.76) | 21.47 (5.55) | / | $39ms^{\dagger}$ | $15.6\times^{\dagger}$ |
| FT [6] (rescoring 10) | 18.66 (4.79) | 22.41 (4.61) | / | $79ms^{\dagger}$ | $7.68\times^{\dagger}$ |
| FT [6] (rescoring 100) | 19.17 (4.28) | 23.20 (3.82) | / | $257ms^{\dagger}$ | $2.36\times^{\dagger}$ |
| IR [9] (adaptive refinement) | 21.54 (3.03) | 25.43 (3.04) | / | / | $2.39\times^{\dagger}$ |
| LT [15] | 19.80 (7.50) | / | / | $105ms^{\dagger}$ | / |
| LT [15] (rescoring 10) | 21.00 (6.30) | / | / | / | / |
| LT [15] (rescoring 100) | 22.50 (4.80) | / | / | / | / |
| CTC [13] | 17.68 (5.77) | 19.80 (7.22) | / | / | $3.42\times^{\dagger}$ |
| ENAT-P [29] | 20.26 (7.15) | 23.23 (8.06) | 25.09 (7.46) | $25ms^{\dagger}$ | $24.3\times^{\dagger}$ |
| ENAT-P [29] (rescoring 9) | 23.22 (4.19) | 26.67 (4.62) | 28.60 (3.95) | $50ms^{\dagger}$ | $12.1\times^{\dagger}$ |
| ENAT-E [29] | 20.65 (6.76) | 23.02 (8.27) | 24.13 (8.42) | $24ms^{\dagger}$ | $25.3\times^{\dagger}$ |
| ENAT-E [29] (rescoring 9) | 24.28 (3.13) | 26.10 (5.19) | 27.30 (5.25) | $49ms^{\dagger}$ | $12.4\times^{\dagger}$ |
| NAT-REG [8] | 20.65 (6.65) | 24.77 (6.52) | 23.89 (9.63) | $22ms^{\dagger}$ | $27.6\times^{\dagger}$ |
| NAT-REG [8] (rescoring 9) | 24.61 (2.69) | 28.90 (2.39) | 28.04 (5.48) | $40ms^{\dagger}$ | $15.1\times^{\dagger}$ |
| VQ-VAE [16] (compress $8\times$) | 26.70 (1.40) | / | / | $81ms^{\dagger}$ | $4.08\times^{\dagger}$ |
| VQ-VAE [16] (compress $16\times$) | 25.40 (2.70) | / | / | $58ms^{\dagger}$ | $5.71\times^{\dagger}$ |
| *Non-autoregressive models (Ours)* | | | | | |
| NART | 20.27 (7.14) | 22.02 (9.27) | 23.04 (10.22) | $26ms^{\ddagger}$ | $14.9\times^{\ddagger}$ |
| NART (rescoring 9) | 24.22 (3.19) | 26.21 (5.08) | 26.79 (6.47) | $50ms^{\ddagger}$ | $7.74\times^{\ddagger}$ |
| NART (rescoring 19) | 24.99 (2.42) | 26.60 (4.69) | 27.36 (5.90) | $74ms^{\ddagger}$ | $5.22\times^{\ddagger}$ |
| NART-CRF | 23.32 (4.09) | 25.75 (5.54) | 26.39 (6.87) | $35ms^{\ddagger}$ | $11.1\times^{\ddagger}$ |
| NART-CRF (rescoring 9) | 26.04 (1.37) | 28.88 (2.41) | 29.21 (4.05) | $60ms^{\ddagger}$ | $6.45\times^{\ddagger}$ |
| NART-CRF (rescoring 19) | 26.68 (0.73) | 29.26 (2.03) | 29.55 (3.71) | $87ms^{\ddagger}$ | $4.45\times^{\ddagger}$ |
| NART-DCRF | **23.44 (3.97)** | **27.22 (4.07)** | **27.44 (5.82)** | $37ms^{\ddagger}$ | $10.4\times^{\ddagger}$ |
| NART-DCRF (rescoring 9) | **26.07 (1.34)** | **29.68 (1.61)** | **29.99 (3.27)** | $63ms^{\ddagger}$ | $6.14\times^{\ddagger}$ |
| NART-DCRF (rescoring 19) | **26.80 (0.61)** | **30.04 (1.25)** | **30.36 (2.90)** | $88ms^{\ddagger}$ | $4.39\times^{\ddagger}$ |

For the WMT14 dataset, we use the default network architecture of the original `base` Transformer [1], which consists of a 6-layer encoder and 6-layer decoder. The size of hidden states $d_{model}$ is set to 512. Considering that IWSLT14 is a relatively smaller dataset comparing to WMT14, we use a smaller architecture for IWSLT14, which consists of a 5-layer encoder, and a 5-layer decoder. The size of hidden states $d_{model}$ is set to 256, and the number of heads is set to 4. For all datasets, we set the size of transition embedding $d_t$ to 32 and the beam size $k$ of beam approximation to 64. Hyperparameter $\lambda$ is set to 0.5 to balance the scale of two loss components.

Following previous works [6], we use sequence-level knowledge distillation [12] during training. Specifically, we train our models on translations produced by a Transformer teacher model. It has been shown to be an effective way to alleviate the multimodality problem in training [6].

Since the CRF-based structured inference module is not parallelizable in training, we initialize our NART-CRF models by warming up from their vanilla NART counterparts to speed up training. We use Adam [30] optimizer and employ label smoothing of value $\epsilon_{ls} = 0.1$ [31] in all experiments. Models for WMT14/IWSLT14 tasks are trained on 4/1 NVIDIA P40 GPUs, respectively. We implement our models based on the open-sourced tensor2tensor library [23].

Table 3: BLEU scores of beam approxiamtion ablation study on WMT En-De.

| CRF beam size $k$ | 1 | 2 | 4 | 8 | 16 | 32 | 64 | 128 | 256 |
|---|---|---|---|---|---|---|---|---|---|
| NART-CRF | 15.10 | 20.67 | 22.54 | 23.04 | 23.22 | 23.26 | 23.32 | 23.33 | **23.38** |
| NART-CRF (resocring 9) | 19.61 | 23.93 | 25.48 | 25.86 | 25.93 | 26.01 | 26.04 | **26.09** | 26.08 |
| NART-CRF (resocring 19) | 20.02 | 25.00 | 26.28 | 26.56 | 26.57 | 26.65 | 26.68 | **26.71** | 26.66 |

## 4.2 Inference

During training, the target sentence is given, so we do not need to predict the target length $T'$. However, during inference, we have to predict the length of the target sentence for each source sentence. Specifically, in this paper, we use the simplest form of target length $T'$, which is a linear function of source length $T$ defined as $T' = T + C$, where $C$ is a constant bias term that can be set according to the overall length statistics of the training data. We also try different target lengths ranging from $(T + C) - B$ to $(T + C) + B$ and obtain multiple translation results with different lengths, where $B$ is the half-width, and then use the ART Transformer as the teacher model to select the best translation from multiple candidate translations during inference.

We set the constant bias term $C$ to 2, -2, 2 for WMT14 En-De, De-En and IWSLT14 De-En datasets respectively, according to the average lengths of different languages in the training sets. We set $B$ to 4/9 and get 9/19 candidate translations for each sentence. For each dataset, we evaluate our model performance with the BLEU score [32]. Following previous works [6, 9, 29, 8], we evaluate the average per-sentence decoding latency on WMT14 En-De test sets with batch size 1 with a single NVIDIA Tesla P100 GPU for the Transformer model and the NART models to measure the speedup of our models. The latencies are obtained by taking average of five runs.

## 4.3 Results and analysis

We evaluate[7] three models described in Section 3: Non-AutoRegressive Transformer baseline (NART), NART with static-transition Conditional Random Fields (NART-CRF), and NART with Dynamic-transition Conditional Random Fields (NART-DCRF). We also compare the proposed models with other ART or NART models, where LSTM-based model [2, 27], CNN-based model [5, 28], and Transformer [1] are autoregressive models; FerTility based (FT) NART model [6], deterministic Iterative Refinement (IR) model [9], Latent Transformer (LT) [15], NART model with Connectionist Temporal Classification (CTC) [13], Enhanced Non-Autoregressive Transformer (ENAT) [29], Regularized Non-Autoregressive Transformer (NAT-REG) [8], and Vector Quantized Variational AutoEncoders (VQ-VAE) [16] are non-autoregressive models.

Table 2 shows the BLEU scores on different datasets and the inference latency of our models and the baselines. The proposed NART-CRF/NART-DCRF models achieve state-of-the-art performance with significant improvements over previous proposed non-autoregressive models across various datasets and even outperform two strong autoregressive models (LSTM-based and CNN-based) on WMT En-De dataset.

Specifically, the NART-DCRF model outperforms the fertility-based NART model with 5.75/7.41 and 5.75/7.27 BLEU score improvements on WMT En-De and De-En tasks in similar settings, and outperforms our own NART baseline with 3.17/1.85/1.81 and 5.20/3.47/3.44 BLEU score improvements on WMT En-De and De-En tasks in the same settings. It is even comparable to its ART Transformer teacher model. To the best of our knowledge, it is the first time that the performance gap of ART and NART is narrowed to **0.61 BLEU** on WMT En-De task. Apart from the translation accuracy, our NART-CRF/NART-DCRF model achieves a speedup of 11.1/10.4 (greedy decoding) or 4.45/4.39 (teacher rescoring) over the ART counterpart.

The proposed dynamic transition technique boosts the performance of the NART-CRF model by 0.12/0.03/0.12, 1.47/0.80/0.78, and 1.05/0.78/0.81 BLEU score on WMT En-De, De-En and IWSLT De-En tasks respectively. We can see that the gain is smaller on the En-De translation task. This may be due to language-specific properties of German and English.

An interesting question in our model design is how well the beam approximation fits the full CRF transition matrix. We conduct an ablation study of our NART-CRF model on WMT En-De task and the results are shown in Table 3. The model is trained with CRF beam size $k = 64$ and evaluated with different CRF beam size and rescoring candidates. We can see that $k = 16$ has already provided a quite good approximation, as further increasing $k$ does not bring much gain. This validates the effectiveness of our proposed beam approximation technique.

## 5 Conclusion and Future Work

Non-autoregressive sequence models have achieved impressive inference speedup but suffer from decoding inconsistency problem, and thus performs poorly compared to autoregressive sequence models. In this paper, we propose a novel framework to bridge the performance gap between non-autoregressive and autoregressive sequence models. Specifically, we use linear-chain Conditional Random Fields (CRF) to model the co-occurrence relationship between adjacent words during the decoding. We design two effective approximation methods to tackle the issue of the large vocabulary size, and further propose a dynamic transition technique to model positional contexts in the CRF. The results significantly outperform previous non-autoregressive baselines on WMT14 En-De and De-En datasets and achieve comparable performance to the autoregressive counterparts.

In the future, we plan to utilize other existing techniques for our NART-CRF models to further bridge the gap between non-autoregressive and autoregressive sequence models. Besides, although the rescoring process is also parallelized, it severely increases the inference latency, as can be seen in Table 2. An additional module that can accurately predict the target length might be useful. As our major contribution in this paper is to model richer structural dependency in the non-autoregressive decoder, we leave this for future work.

## Footnotes

*Equal contribution. Work was done while visiting Microsoft Research Asia.

[1]The reproducible code can be found at `https://github.com/Edward-Sun/structured-nart`

[2]The transition is calculated in the log space. See `https://github.com/tensorflow/tensorflow/tree/master/tensorflow/contrib/crf` for detailed implementation.

[3]https://wit3.fbk.eu/

[4]http://statmt.org/wmt14/translation-task.html

[6]In Table 2, $\ddagger$ and $\dagger$ indicate that the latency and speedup rate is measured on our own platform or by previous works, respectively. Please note that both of them may be evaluated under different hardware settings and it may not be fair to directly compare them.

[7]We follow common practice in previous works to make a fair comparison. Specifically, we use tokenized case-sensitive BLEU for WMT datasets and case-insensitive BLEU for IWSLT datasets.

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
