[Reviews · NeurIPS 2019]

Reviewer 1



The paper proposes to boost translation quality of a non-autoregressive (NART) neural machine translation system through a conditional random field (CRF) that is attached to the decoder. The CRF reduces the translation quality drop compared to autoregressive neural translation systems by imposing a bigram-language model like structure onto the decoder that helps to alleviate the strong independence assumption that NART architectures entail. The CRF is jointly trained with all other parameters of the neural network. Experiments conducted on WMT14 and IWSLT14 En-De and De-En tasks are reported to yield improvements of more than 6 BLEU points over their corresponding baselines. By augmenting the decoder with a Markov-order 1 CRF, the resulting network is strictly speaking no longer a non-autoregressive system. The CRF has similar qualities as using a bigram-language model, and even if the increase in latency at inference time is small, one may yield similar quality improvements with only marginal latency increase by choosing one of the many other autoregressive components as the last decoder layer. (Even collecting the bag of top-k scored tokens at each target sentence position and conducting a fast beam-search using a trigram language model may already give similar improvements.) The paper does not describe on which hardware the latency measurements were taken, and there is also no explanation of how the rescoring experiments reported in Table 2 were conducted. L34: The citations on previous studies is incomplete. E.g., the work by Jason Lee, Elman Mansimov and Kyunghyun Cho on "Deterministic Non-Autoregressive Neural Sequence Modeling by Iterative Refinement" is missing. L38: In Equation~(2), the variable z is unbound, and its explanation as "a well-designed input z" is insufficient for its understanding. L40: The examples shown in Table 1 refer to a specific NART model, and the authors present these examples as if any NART model would exhibit the same type of translation errors, which is clearly not the case. For better clarity, the authors should provide (or reference) a description of the specific architecture and hyper-parameter choice of their NART model from which they derived their examples. L44: The use of the terms "multimodality" and "multimodal distribution" seems inappropriate and somewhat of a misnomer in this context: There is no indication that the target distribution has more modes ("peaks") than can be captured by the network. The root cause merely seems to be the independence assumption. The same goes for L82 L83 and L204. Maybe the authors meant multinomial (?). L59: It's best to give a reference to those subsections that will describe the low-rank approximation and the beam approximation. L140: Z(x) is not a normalization *constant* but a normalization factor (aka. as partition function). L214 The approach to predict the target length T' merely as the source length T offset by a constant C seems implausible: Particularly for a language such as German, where compounds occur more frequently compared to English, one would expect a linear relationship between T and T'. Would it not be more plausible to make T' a constant and use padding tokens to fill up the target sentences? L222: The authors should make clear whether they compute case-sensitive or case-insensitive BLEU scores. Judging from the examples given in Table 1, I conject that a case-insensitive BLEU score evaluation was used. A case-insensitive evaluation artificially inflates BLEU scores though, whereas the baseline numbers in Table 2 that are cited from literature report case-sensitive BLEU scores and therefore tend to be nominally lower. The paper does not describe how the rescoring results reported in Table 2 were set up nor what the numbers 10, 100, 9, and 19 refer to.

Reviewer 2



The paper proposes to view MT as a sequence labeling problem modeled with a linear chain CRF, noting that such an approach becomes feasible if the length of the target output is fixed or predicted (as is common in non-autoregressive MT). The authors use a more or less standard transformer encoder-decoder architecture, but the decoder is non-autogressive and simply consumes a fixed number of padding tokens, and the log probability of the sequence is modeled with a CRF, which makes use of the transformer outputs at each time-step. Experimentally, the authors show that they can outperform many recent non-autoregressive MT baselines, while attaining comparable speedups. Originality: as noted above, this does appear to be an interesting and rather original idea, at least for neural MT. I think the main promise of this approach is in exact decoding, though the authors do not investigate this much. Quality and Clarity: Though the paper is easy to follow, I think the presentation could be improved in several respects: - I think it's a little strange to refer to the proposed method as non-autoregressive; it is autoregressive, though it uses only the previous label/token as its history. - Equations (2) and (3) should be corrected so there is no p(z|x), since z (which is the input of padding tokens) is not random and is not modeled. Similarly, if p(T'|x) is random (which it doesn't appear to be) the left-hand-sides should be changed to p(y, T' |x). - I think the discussion of the proposed method's runtime on lines 178-181 needs to be longer and perhaps formalized a bit more: in particular, the authors should justify why they view their proposed approach as being O(n k^2), and more pressingly, what they view the complexity to be of decoding under the models with which they compare. For instance, what do they view as the decoding complexity of an RNN-based decoder (perhaps with no attention)? Experimentally, the authors compare with a large number of recent baselines, which is very impressive. However, I believe some of the baselines could be improved. In particular, the Transformer baseline appears to use a beam size of 4, which will slow it down. It would be good to see its performance with a beam of size 1. Even better, training a Transformer model on the beam-searched outputs of a teacher Transformer model (i.e., with knowledge distilliation) can often lead to improved performance even with greedy decoding; note that this is the most fair comparison, since the non-autoregressive models are also trained from a teacher Transformer model. Furthermore, the authors do not include timing information for the RNN decoder, which should also be linear in the length of the output sequence. (Attention to the source complicates this a bit, though there are recent models (e.g., Press and Smith (2018)) that get good performance without it). Update: Thanks for your response. I'm increasing my score to a 7 in light of the response, especially given the distilled greedy Transformer results.

Reviewer 3



My main concern with this paper is authors calling their approach non-autoregressive. Running forward backward algo in CRF breaks that conditional independence assumption among predicted tokens which makes the proposed approach autoregressive. Although I would agree with authors that decoding with CRF is much more lightweight and is faster compared to decoding with autoregressive Transformer, where output has to be fed back in as an input into a deep sequence model. I would suggest authors using word "fast" neural machine translation instead of "non-autoregressive" neural machine translation in the paper. Regarding model itself, I would also like to see an ablation study on the effect of vanilla non-autoregressive loss as well use of different target lengths on the final performance in Table 2. Apart from my points above overall I believe it is a well executed paper that introduces several techniques to make CRF + Deep Neural Nets applicable for Neural Machine Translation and I recommend its acceptance.

[Author Response · NeurIPS 2019]

We thank all reviewers for the valuable comments. First, we respond to two common concerns raised by the reviewers, and then answer other questions raised by each reviewer.

---

**[Comment: By augmenting the decoder with a CRF, the resulting network is no longer non-autoregressive.]**

We will use a new title "Structured Decoding for Fast Machine Translation", as suggested by Reviewer #5. We will also improve our presentation in other related parts of our paper. However, we still want to emphasize our speedup comparing to vanilla autoregressive models.

**[Comment: In Equation 2 and 3, the variable z is unbound, and its explanation as "a well-designed input z" is insufficient for its understanding.]**

This is a general formula for non-autoregressive machine translation. In our NART, $z$ is deterministic, but in other models such as (Gu et al., 2017) or (Roy et al. 2018), $z$ is stochastic. For example, in (Gu et al.), $z$ is a fertility-related copy of source tokens, while in (Roy et. al), $z$ is a sequence of autoregressively generated discrete latent variables. We will make our presentation more clear in the final version. Specifically, we will fix the typo in Equation 3: $n$ should be $T'$ over the $\Sigma$ mark.

---

**To Reviewer #3** Thanks for the detailed review! Regarding your comments: **(1)** *"A fast beam-search using a tri-gram language model may already give similar improvements."* This is a possible approach to extending our current practice. However, we would like to emphasize that although a linear-chain CRF is similar to a bi-gram language model, it has many other advantages such as **end-to-end** training with neural networks, as well as **exact inference** over the labeling/translation, as pointed out by Reviewer #4. Moreover, the autoregressive part should be simple enough to keep small overhead. **(2)** We describe the hardware setting at L224. **(3)** Since the target length is fixed, we try different target lengths ranging from $(T + C) - B$ to $(T + C) + B$ (at L216, Sec 4.2). Therefore, we set $B$ as 4 and 9 for rescoring 9 and 19 candidates. Rescoring 9 is for a fair comparison with previous work, while rescoring 19 is to explore the limits of our model. We will improve our presentation about this in the final version and include descriptions of how other models rescore their candidates. **(4 - L40)** Note that the examples in (Gu et al., 2017) also show this type of translation errors. This inconsistent pattern is common in NART literature and we will add references about this type of translation error. The examples shown in Table 1 are from our implemented models (as described in Sec 4 of the paper). **(5 - L214)** In order to show the effectiveness of the structured inference module, we choose the simplest target length prediction model. Please note that the precision of the predicted target length becomes less important when we rescore more candidates. Thank you for your suggestion. Actually, we have already noticed this problem in the future work section (L272) and plan to improve the target length prediction module in our future work. **(6 - L222)** We follow common practice in previous works to make a fair comparison. Specifically, we use tokenized case-sensitive BLEU for WMT datasets and case-insensitive BLEU for IWSLT datasets. We will include this in the final paper. **(7)** Regarding other comments, please refer to the general response sections for L38, and we will fix the issues at L34, L59 and L140 in the final version. We appreciate your detailed review comments and hope the response address your concerns.

**To Reviewer #4** Thanks for your positive comments! Regarding your suggestions: **(1)** *"I think the main promise of this approach is in exact decoding, though the authors do not investigate this much."* Thank you for your comment. We will provide more discussion about this direction in the final version. **(2)** $O(nk^2)$ complexity is the FLOPs (mul-adds) of the unparallelizable part of CRF module. In comparison, the time complexity of an one-layer RNN-based decoder without attention mechanism is $O(nh^2)$, where $n$ is sequence length, $h$ is the number of hidden units and $h$ is usually several times larger than $k$. We will provide more discussion about the time complexity in the final version. **(3)** According to your suggestion, we conducted a set of experiments and here are the results (LSTM-based is from a 8-layer modification of `lstm_attention_base` in `tensor2tensor` library, which has similar model size with `transformer_base`):

| **Models** | WMT14 En-De | Latency | Speedup |
|---|---|---|---|
| LSTM-based (beam size = 1) | / | 2031.6*ms* | / |
| Transformer (distilled, beam size = 1) | 26.48 | 240*ms* | 1.61$\times$ |

**To Reviewer #5** Thanks for your positive comments! Please refer the previous general response section about some of your concerns. **(1)** For the use of different target lengths, we tried different $C$ and find that the current setting yields the best performance. **(2)** The results of ablation study on the effect the vanilla non-autoregressive loss:

| $\lambda$ | 0.0 | 0.1 | 0.5 | 1.0 |
|---|---|---|---|---|
| WMT14 En-De (NART-CRF, no rescoring) | fail | 22.42 | **23.32** | 22.59 |

[Meta-Review · NeurIPS 2019]

The reviewers and I find the paper interesting, especially because such a simple approach performs favorably in comparison with non-autoregressive and expressive autoregressive models for machine translation. I recommend acceptance as a poster given that the reviewers raise several concerns about the original manuscript. I ask the authors to change the title as agreed in the rebuttal by using terms such as low-latency, fast, etc. It seems that the paper uses approximate partition function for training which is is not explained in details. The theoretical properties of such an approximation may be interesting to study. The submission should cite and discuss relevant previous work on combining neural networks with CRF for sequence labeling such as Andor et al., 2016: https://www.aclweb.org/anthology/P16-1231 and Collobert et al. 2011: http://www.jmlr.org/papers/volume12/collobert11a/collobert11a.pdf.